# Investigating the Role of Maintenance TMS Protocols for Major Depression: Systematic Review and Future Perspectives for Personalized Interventions

**DOI:** 10.3390/jpm13040697

**Published:** 2023-04-21

**Authors:** Giacomo d’Andrea, Gianluca Mancusi, Maria Chiara Santovito, Carlotta Marrangone, Fabrizio Martino, Mario Santorelli, Andrea Miuli, Francesco Di Carlo, Maria Salvina Signorelli, Massimo Clerici, Mauro Pettorruso, Giovanni Martinotti

**Affiliations:** 1Department of Neurosciences, Imaging and Clinical Sciences, Università degli Studi G. D’Annunzio, 66100 Chieti, Italy; giacomo.dandrea1993@gmail.com (G.d.); g.mancusi@hotmail.com (G.M.); mariachiarasantovito@ymail.com (M.C.S.); carlotta.marrangone@gmail.com (C.M.); martino.fabrizio1@fastwebnet.it (F.M.); andreamiuli@live.it (A.M.); francesco.dic@hotmail.it (F.D.C.); giovanni.martinotti@gmail.com (G.M.); 2Psychiatric Residency Training Program, School of Medicine and Surgery, University of Milano Bicocca, 20900 Monza, Italy; santorelli.m@gmail.com (M.S.); massimo.clerici@unimib.it (M.C.); 3Psychiatry Unit, Department of Clinical and Experimental Medicine, University of Catania, 95123 Catania, Italy; maria.signorelli@unict.it; 4Psychopharmacology, Drug Misuse and Novel Psychoactive Substances Research Unit, School of Life and Medical Sciences, University of Hertfordshire, Hatfield AL10 9AB, UK

**Keywords:** repetitive transcranial magnetic stimulation, maintenance, major depressive disorder, treatment-resistant depression, neuromodulation, rescue TMS, tapering TMS, continuation TMS, cluster TMS, personalized medicine

## Abstract

Repetitive Transcranial Magnetic Stimulation (rTMS) has been approved by the FDA as an effective intervention for Treatment-Resistant Depression (TRD). However, there is little evidence about maintenance protocol necessity. The aim of this systematic review is to identify, characterize, and evaluate the current maintenance TMS protocols for MDD and TRD patients who have received acute treatment. A literature search was conducted following the PRISMA guidelines of 2015 on PubMed, Scopus, and Web of Science databases for publications up to March 2022. Fourteen articles were included. High protocol heterogeneity was observed. Most studies highlighted significant efficacy of maintenance protocols in decreasing relapse risk, suggesting that administering two or fewer stimulations per month is ineffective in sustaining an antidepressant effect or in reducing the risk of relapse in responder patients. The risk of relapse was most pronounced after five months from the acute treatment. Maintenance TMS appears to be a resourceful strategy to maintain acute antidepressant treatment effects, significantly reducing relapse risk. The ease of administering and the ability to monitor treatment adherence should be considered when evaluating the future use of maintenance TMS protocols. Further studies are needed to clarify the clinical relevance of overlapping acute TMS effects with maintenance protocols and to evaluate their long-term effectiveness.

## 1. Introduction

Major Depressive Disorder (MDD) is a widespread mental health condition that affects approximately 280 million people globally, with a prevalence of 5.0% among adults and higher rates among individuals over 60 years of age [1]. It is typically characterized by depressed mood and anhedonia, with a significant impact on various aspects of an individual’s overall functioning, including psychological, cognitive, physiological, occupational, and social dimensions, often becoming a life-threatening condition due to comorbidity, low self-care, and suicidal behaviors. These clinical features make MDD a serious illness with a heavy impact on global health. MDD is often characterized by a clinical pattern of recurrence and chronicity, therefore, being a lifelong clinical condition to deal with. Lifetime risk of relapse after the first MDD episode is estimated to be around 60%, growing to 70% after two MDD episodes, and up to 90% in patients who experienced more than two episodes [2]. Moreover, each relapse increases the chances of chronicity and the development of treatment resistance [3]. From 30% to 50% of MDD patients do not respond to first-line psychopharmacological treatments [4], and rates of remission decrease to 60% after four antidepressant failures [3]. In this context, Treatment-Resistant Depression (TRD) is defined as the absence of a clinical response after at least two antidepressant treatments administered for adequate doses and for at least 4–6 weeks [5]. TRD is a serious global health issue with a double risk of hospitalization and seven times higher suicide risk compared to treatment-responder depressed patients [6]. Strategies with worldwide consensus on TRD management include different approaches: continuation of antidepressant therapy, optimizing dose and time; changing antidepressant drug, switching to another pharmacological class; augmentation with other psychopharmacological agents (mood stabilizer, second-generation antipsychotics) [7]. However, the use of multiple and complex psychopharmacological treatments to overcome treatment-resistance can lead to lower adherence, thus determining significant increases in relapse or experiencing more severe depressive episodes [8,9]. Furthermore, antidepressant therapy may manifest individual blood fluctuation depending on the specific metabolic pattern of a patient. In this context, common MDD and TRD psychopharmacological treatments could not be fully reliable strategies.

Repetitive Transcranial Magnetic Stimulation (rTMS) is a non-invasive brain stimulation technique approved by the FDA as an effective intervention for TRD, largely supported by the current literature, indicating response rates of 40–50% and remission rates of 25–30% [10]. TMS treatment represents a reliable and reproducible therapy that can be administered at known dosages and predictable neural effects, with few side effects and a low risk of manic switch [11]. Thanks to its easy use and the aforementioned low risk of side effects, rTMS could be used as an augmentation therapy, thus guaranteeing a more comprehensive management of symptoms [12]. Conventional acute protocols for MDD last 4–6 weeks and include high-frequency (HF) activating stimulations on the left DLPFC, low-frequency (LF) inhibitory stimulations on the right DLPFC, or a combination of both [13]. Its mechanism of action relies on its capability of activating specific brain cortical regions (i.e., dorsolateral prefrontal cortex, DLPFC) whose activity is altered in MDD [14], inducing changes in neural structure and functionality. In particular, several studies reported improvements in cortical thickness and neuroplasticity corresponding to neurobiological phenomena, such as long-term potentiation (LTP) and BDNF modulation [15,16].

In a neurobiological framework, rTMS seems to act directly on aberrant connections between prefrontal areas and the anterior cingulate cortex [17], thus restoring reward network activity and hedonic function [18].

A recent study on this matter pointed out that a significant portion of patients (up to 63%) does not respond to acute rTMS protocols immediately but rather shows a delayed response [19]. This finding suggests that the mechanisms underlying magnetic stimulation are complex and involve changes that occur over time. 

The duration of the effects of acute TMS treatment on responder patients has been estimated to be approximately 5 months, at which point the relapse rate increases to 20% [20,21]. Given the high frequency of recurrent depressive episodes in the natural history of MDD and TRD and the significant proportion of patients who experience relapses after acute TMS treatment, maintenance TMS protocols may be able to extend or enhance the neuroplastic and functional changes achieved after acute treatment, thus possibly reducing recurrence in depressive disorders. 

Nevertheless, the role of maintenance TMS has not been fully established (these protocols, for instance, are not included in the international treatment guidelines for depressive disorders), with high heterogeneity in administration between studies. Furthermore, neuromodulation therapies require commitment from both patients and healthcare systems, and thus, solid evidence is required to prove the benefits of this therapeutic strategy before its applications in clinical practice. Therefore, the purpose of this systematic review is to identify, characterize, and synthesize the current evidence on maintenance TMS protocols for patients with MDD or TRD to provide clear guidance to clinicians on the impact of conducting these protocols in reducing the risk of relapse and recurrences in depressive disorders.

## 2. Materials and Methods

We reviewed English language original articles (open-label/double-blind trials, prospective/retrospective observational studies, case series, case reports) on the use of TMS (rTMS; deep TMS, DTMS) conducted on either MDD or TRD subjects.

### 2.1. Systematic Review Procedures

A systematic electronic search was performed on the 25 March 2022 on PubMed, Scopus, and Web of Science (WoS) databases. The following search strategies have been used, respectively, in PubMed and WoS (TMS OR Transcranial Magnetic Stimulation) AND (MDD OR Major Depressive Disorder OR Unipolar Depression OR Depression) AND (follow-up OR follow up OR maintenance OR continuation) NOT (review OR meta-analysis OR animal OR in vitro); in Scopus: (TITLE-ABS-KEY (tms) OR TITLE-ABS-KEY (transcranial AND magnetic AND stimulation) AND TITLE-ABS-KEY (mdd) OR TITLE-ABS-KEY (major AND depressive AND disorder) OR TITLE-ABS-KEY (unipolar AND depression) OR TITLE-ABS-KEY (depression) AND TITLE-ABS-KEY (follow-up) OR TITLE-ABS-KEY (follow AND up) OR TITLE-ABS-KEY (maintenance) OR TITLE-ABS-KEY (continuation) AND NOT TITLE-ABS-KEY (review) OR TITLE-ABS-KEY (meta-analysis) OR TITLE-ABS-KEY (animal) OR TITLE-ABS-KEY (in AND vitro)).

The systematic review was structured in accordance with the PRISMA (see Figure 1) [22] and PROSPERO guidelines [23]. Identified studies were assessed at title/abstract and full-text screening against eligibility criteria.

### 2.2. Data Synthesis Strategy

The search of results was carried out individually by four investigators (MCS, GMa, CM, and FM) and supervised by AM, GdA, FDC, and MS; doubtful cases were discussed by GM and MP. The exclusion criteria for both selection phases were: (1) non-original studies (e.g., review, commentary, editorial, book chapter); (2) non full-text articles (e.g., meeting abstract); (3) language other than English; (4) animal/in vitro studies; (5) articles not dealing with TMS treatment; (6) articles not dealing with maintenance protocol (7) articles not dealing with MDD. Removing duplicate articles (*n* = 502) from a total of 1342 papers (PubMed = 432; Scopus = 507; WoS = 403), 840 records have been screened, and, among these, 671 were not relevant to the subject reading title and abstract (animal/in vitro studies, articles not dealing with TMS treatment, maintenance protocol or MDD), 32 were not written in English, and 93 were non-original articles (e.g., review, metanalysis, commentary, letter to the editor without data available, book chapter). Of the 44 full-text articles assessed for eligibility, 30 did not match the inclusion criteria for our review. Finally, 14 articles were included (Figure 1). All these research methods were approved by PROSPERO (identification code CRD42022321142).

### 2.3. Assessment of Risk of Bias

Quality assessment was performed by two members of the team (MS and AM), with discrepancies that could not be resolved by discussion being solved by consulting additional investigators (MP, GM). The risk of bias was assessed with risk of bias 2 tool (ROB-2) for RCTs, risk of bias in non-randomized studies of interventions 1 (ROBIN-1) for non-RCTs, and “The Critical appraisal checklists for case series and case report of Joanna Briggs Institute” for case reports and case series. The risk of bias assessment of included RCT and N-RCT studies consists of three with a severe risk of bias, four with a moderate risk and, three with a low risk of bias, with the most common error in the “measurement of outcomes” area (see Appendix A).

### 2.4. Data Extraction and Qualitative Synthesis

The full-text articles of the 14 studies eligible for data extraction were independently assessed by two independent reviewers (GdA and GMa), with discrepancies that could not be resolved being solved by consulting additional investigators (MP, GM). The data extracted included: bibliographic details; study design; sample size and diagnosis; acute intervention duration; targeted area; coil position; for acute protocols: frequency, intensity, session duration, number of pulses, number of total sessions; distance between acute and maintenance protocols; for the maintenance protocols: duration and number of stimulations per week, frequency, intensity, number of pulses, number of total sessions; assessments methods; results. As an outcome measure, we considered the variation among psychometric scales, together with response and remission rates during the treatment protocol. Furthermore, relapse rates of MDEs, if present, were reported.

## 3. Results

Findings are described in detail and organized related to alphabetical order, type of study, and population involved (MDD or TRD) (see Table 1 and Table 2). From the total of 14 articles, 3 were randomized sham-controlled trials (RCTs), 8 were open-label studies, 2 were case reports, and 1 was a case series. The sample comprised 461 patients suffering from TRD and 386 from MDD.

### 3.1. Randomized Control Trials

Of the three RCTs [24,25,26], two [24,25] are characterized by maintenance protocols following acute treatment with rTMS, while in one RCT [26], monthly clustered rTMS maintenance was administered to MDD subjects after achieving partial or full remission from the MDE through oral antidepressant therapy. In the aforementioned study [26], MDD remitters were randomly assigned to an rTMS (*n* = 91), antidepressant (*n* = 108), or combined (rTMS + ADP, *n* = 82) treatment group for 12 months; starting a 52-week maintenance protocol of HF rTMS stimulating LDLPFC results in a significantly reduced risk of relapse compared to the sole antidepressant treatment [26] supporting rTMS efficacy of prevent MDE relapse compared to an active comparator. Coherently with the previous article, Levkovitz and colleagues found that a 12-week maintenance protocol of twice-a-week HF dTMS over LDLPFC ensured higher levels of response and remission in comparison to sham stimulation, and these differences were stable for the whole maintenance phase [25]. Conversely, another RCT failed to find significant long-term differences in terms of antidepressant response between active and sham groups in 35 TRD subjects: in this 44-week HF rTMS maintenance protocol, a significant reduction of depressive symptoms in the active group compared to sham was detected between the first and the fourth month, while no significant differences were found from the fifth month to the endpoint [24].

### 3.2. Open-Label Studies

From the eight open-label studies [27,28,29,30,31,32,33,34], one used the right DLPFC using LF rTMS as the targeted area [28], one used mixed protocols, stimulating left DLPFC, right DLPFC, and bilateral in three different subgroups [27], one used bilateral protocols (HF rTMS on the left DLPFC, LF rTMS on the right DLPFC) [33], while the other five studies [29,30,31,32,34] used all HF rTMS stimulating LDLPFC.

Different maintenance protocols were used in these open-label studies. Two studies reported a twice-a-week maintenance stimulation protocol, one for a total duration of 8 weeks [29] and the other for 12 weeks [34]. Both indicate a higher percentage of responders at the end of the maintenance phase (81.12% for Harel et al. [29] and 72.7% for Yip et al. [34] respectively), although no data about the relapse rates are reported. In particular, the study conducted by Yip and colleagues reported the efficacy of a double-blinded maintenance rTMS protocol in TRD subjects who were not previously responders to an acute rTMS treatment [34].

Two other studies [28,33] reported tapering rTMS maintenance protocols, with a progressive reduction in the session numbers over time. In Haesebaert et al. [28], 66 TRD patients, who responded to rTMS (*n* = 25), venlafaxine (*n* = 22), or a combination of both treatments (*n* = 19), continued to receive the treatment that led to a response as a maintenance treatment over 12 months, with no different efficacies in relapse prevention and the maintenance of remission in TRD patients. On the other hand, Richieri et al. [33] reported a significantly lower relapse rate in TRD patients among responders treated with maintenance rTMS compared to no additional rTMS treatment (37.8% vs. 81.8%) [33].

Three OL studies reported once monthly rTMS maintenance protocols [30,31,32]. Pridmore and colleagues reported data from a naturalistic, open-label observational 10-month maintenance rTMS study in two different articles [31,32]. In the first article [32], data were acquired from 39 TRD patients who had experienced relapse within 3 months following acute TMS: subjects underwent 20 weeks of maintenance HF once monthly (five rTMS series over 3 or 5 days per month), rTMS stimulating program over left DLFPC, and at the end of the 20 weeks, 79% were in remission from the MDE [32]. In the second one, 14 TRD patients underwent the same rTMS maintenance protocol for 52 weeks, and 12 subjects (85%) were in remission during and after the TMS maintenance period, with a relapse rate of 15% [31]. Another study compared a 40-week once monthly HF rTMS maintenance protocol with clinical observation in 49 medication-free MDD patients who meet the criteria for a clinical response after an acute rTMS protocol, showing no significant group differences on any outcome measure and thus suggesting that a single session monthly of rTMS is not sufficient to avoid relapse [30].

Moreover, another study analyzed the efficacy of monthly maintenance rTMS in 35 TRD patients, with 5 sessions monthly divided into three different days for a mean duration of 12 ± 9.7 months [27]. These subjects were divided into four different subgroups: 14 subjects underwent HF rTMS on the left DLPFC, 12 subjects LF on the right DLPFC, 6 subjects experienced bilateral LF rTMS, while 3 subjects underwent bilateral HF/LF rTMS over left/right DLPFC. Comprehensively, 25 subjects experienced a relapse, with 10 subjects, subsequently responding to a new active rTMS protocol (rescue rTMS), while 10 subjects (28.5%) were in remission during all study periods.

### 3.3. Case Reports and Case Series

Two case reports and one case series reported data on maintenance TMS protocols [35,36,37]. One study reported the case of a TRD patient who underwent an HF acute rTMS protocol over left DLPFC and then experienced three different relapse episodes, all treated successfully with two different rTMS protocols (first relapse: 5 sessions over one month, second and third relapses: 20 sessions over two months) [36]. Furthermore, another study reports the successful treatment of a MDD patient with a 12-month HF rTMS maintenance protocol (5 sessions of daily rTMS monthly, for a total of 60 sessions) without experiencing relapses [37]. In a case series of 10 TRD patients, O’Reardon and colleagues retrospectively reviewed the charts of 10 TRD patients treated from 6 months to 6 years with maintenance rTMS protocols, typically 1 or 2 sessions per week, and observed a marked and sustained benefit in 70%, with 3 subjects in monotherapy with rTMS [35].

## 4. Discussion

TMS is a reliable and effective intervention to treat MDD and TRD. Acute TMS treatment protocols are mainly standardized, prevailing the one-a-month HF protocol over left DLPFC (usually 20 sessions), which appears to have a reliable efficacy profile. On the other hand, a global consensus lacks how a maintenance protocol should be carried on. The present article reviewed the current evidence about TMS maintenance protocols for MDD and TRD.

Most of the included studies highlighted the significant efficacy of maintenance protocols in decreasing relapse risk. However, an overall view of included articles highlighted a wide heterogeneity in maintenance protocols applied in almost every parameter (i.e., brain region stimulated, stimulation frequency, frequency of sessions, the temporal distance from acute treatment, and duration of maintenance treatment). Table 3 represents the different maintenance protocols observed in the retrieved studies, which can be resumed in four distinct types (tapering, cluster, and continuous and rescue rTMS). Due to this large heterogeneity observed, it is difficult to unequivocally identify which parameters can mostly affect the capacity of maintenance TMS to prevent relapses. However, by contrasting successful studies to those with insignificant or negative outcomes, some conclusions can be drawn.

### 4.1. Target Populations, Stimulation Frequency, and Target Brain Areas

Considering the target population, most of the included studies treated patients who responded to acute TMS treatments. However, in some cases [26,28], patients treated with classical antidepressant therapies were also included, as well as non-responder patients [29]. The heterogeneity in the studied populations may have increased the differences observed in outcomes between maintenance protocols. Despite the considerable heterogeneity in the target population selected for maintenance TMS protocols across various studies, it appears that implementing maintenance protocols in patients who have already responded to acute TMS treatments may exhibit a degree of efficacy.

The type of stimulation frequency and the target area was also different among studies. The most common brain region stimulated was the left DLPFC through HF-activating protocols (i.e., 10–20 Hz). The right DLPFC was also targeted through LF inhibitory protocols (i.e., 1–5 Hz) in two cases [27,28]. This apparently discretionary choice could reflect the lack of superiority evidence of LF- or HF-stimulation protocols over right or left DLPFC in acute depressive episodes [38]. In fact, to date, no superior efficacy has been observed in different RCTs involving left, right, or bilateral DLPFC acute stimulation, and the target area is usually chosen at the researcher’s discretion [38,39]. We may argue that both HF and LF protocols over left/right areas could be successfully used also in maintenance treatments.

### 4.2. Frequency of Maintenance Sessions

Benadhira et al. [24] noticed a conspicuous enhancement in depressive manifestations during the fourth month from treatment beginning, while no disparities in relapses between active and control groups were observed from the fifth to the eighth month. Maintenance in this last period was characterized by only two treatments per month [24]. Haesebaert et al. [28] found no significant difference between the groups in an open-label study comparing remission and relapse rates over different treatment approaches (i.e., rTMS, venlafaxine, or a combination of the two). The number of maintenance sessions was two per month starting from the fourth month for a LF protocol [28]. Philip et al. [30] also found no significant differences between a group following a maintenance protocol with a single monthly TMS session and an observational group, suggesting a single session per month could not be sufficient to prevent recurrence [30]. Intriguingly, all the other studies selected, including protocols with more than two stimulations per month or a total number of sessions greater than 34 [25,26,27,29,31,32,35,36,37], reported a significantly reduced risk of relapse overall, thus effectively sustaining a response status in depressed patients. These observations suggest that administering two or fewer stimulations per month may be ineffective in sustaining an antidepressant effect or in reducing the risk of relapse in responder patients.

Moreover, it is a consolidated notion that concerning acute protocols, the presence of multiple daily stimulations is associated with a higher probability of patient response, according to a dose–response relationship [40]. Similarly, there is evidence indicating that, independently of the length of the TMS treatment, the difference in the antidepressant response is given by the total number of stimulation sessions. This evidence is the basis for the use of accelerated TMS protocols [41]. If we transfer this evidence to maintenance protocols, we may argue that a low number of sessions per day, as well as a reduced number of total monthly sessions, could be associated with a higher risk of inadequate antidepressant action of TMS. Speculatively, this action could be related to the effect of TMS on neuroplasticity, which manifests itself following a prolonged stimulation of the synapse, as indicated by several studies [42]. Additionally, the absence of repeated stimulation at short intervals may not ensure the preservation of the advantageous outcomes of neuroplasticity induced by TMS.

### 4.3. Temporal Distance from Acute to Maintenance Protocol

A crucial aspect to take into account when determining a maintenance protocol is whether it should be temporally distinct from the acute protocol or be a continuation of it. Studies showed that maintenance can start from one week [24,25,29,34,35] to one month [27,28,30,31,32,36,37] after acute treatment. This systematic review did not identify any significant difference in outcomes depending on the amount of time between acute and maintenance protocols. Nonetheless, additional examination is warranted. It has been noted that the antidepressant effects of TMS were not limited to the acute treatment period, but a considerable number of patients (up to 63%) showed a response even weeks or months after. Additionally, post-acute treatment relapse risk was very low in the first month [19]. In this context, maintenance starting before this period and close to acute treatment could overlap with active protocol efficacy, hence suggesting a possible superfluous treatment. Furthermore, results have indicated that a four-week gap between acute and maintenance protocols did not lead to adverse outcomes [31,32,37], thus proposing four weeks as an optimal distance between the two protocols.

### 4.4. Duration of Maintenance Protocol

The maintenance protocols included in this review have a wide difference in duration. Mean maintenance durations spanned from 12 weeks [25,34] to more than one year, except in one case, which reported only one week [36]. Duration appeared to be a crucial, nevertheless widely different, parameter in maintenance protocols. A first point of interest is how long acute TMS treatment effects span over time. Follow-up studies observed a rise in relapse risk after about five months of distance from acute treatment [20,21], suggesting that responder patients could maintain a stable remission of symptoms for some time without the need for maintenance treatment. Studies included in this review with a duration time of 12 or fewer weeks all had positive outcomes [25,34,36], but it is difficult to discern whether the effect was linked mainly to the acute or the maintenance treatment. Harel et al. [29] reported a dramatic increase in response probability after an 18-week maintenance period (from 46.15% after four weeks of acute treatment to 81.12% after maintenance) [29]. As already mentioned before, part of this result could be correlated to the “late-responder” phenomenon [19], but it could also be seen as an “enhancement” of the acute effects. Undoubtedly, studies lasting longer than five months can be deemed as more reliable and less affected by immediate treatment effects. Among these, all but three studies characterized by a single monthly session [24,28,30] showed a significant reduction in relapse risk among responder patients. Further studies are needed to better clarify the clinical relevance of overlapping acute TMS effects with maintenance protocols in the short period after acute treatment.

### 4.5. A Possible Reliable Maintenance Protocol

In this highly heterogeneous context, is it possible to hypothesize a maintenance TMS protocol able to preserve the response rates and reduce the risk of relapse? The few studies conducted up to now appeared to be contradictory and far from forming a consensus, yet the findings in this review may allow us to draw some general conclusions.

With respect to the target area, although the majority of studies targeted the left dorsolateral prefrontal cortex (LDLPFC), there is insufficient evidence to demonstrate that protocols targeting the right dorsolateral prefrontal cortex (RDLPFC) or bilateral stimulation are inferior in efficacy compared to LDLPFC ones. Nevertheless, it seems reasonable to target the same area in both acute and maintenance stimulation, taking into account that TMS induces specific cortical structural and functional changes that directly impact the clinical manifestations of depression [15,16] and should thus be sustained to perpetuate the acute effects.

In view of the delayed effect of acute TMS over the subsequent months [24], it may be reasonable to keep the acute treatment and the maintenance one separated by at least one month as a maintenance regimen in the initial month after the acute TMS may be superfluous in a situation with low relapse risk in responder patients. Furthermore, while research has revealed that a reaction to the acute phase is indicative of a response to maintenance treatment [29], it is still uncertain whether a maintenance treatment during the initial month subsequent to acute TMS may augment the efficacy among non-responder patients.

Findings from this review emphasize the importance of maintaining the protocol for at least five months following the acute treatment phase, as after this period, the risk of relapse is most pronounced [19]. Furthermore, the number of monthly stimulations should exceed two, as suggested by Benadhira et al. [24] and supported by the studies which demonstrated negative outcomes when only one or two monthly treatments were administered [24]. However, it is unclear whether these stimulations should be administered on separate days or on a single stimulation day. Comprehensively, there is no available evidence about the type of maintenance protocol (i.e., cluster rTMS, tapering rTMS, continuous rTMS, or rescue rTMS) able to guarantee higher levels of effectiveness. Nevertheless, cluster and continuous rTMS studies seem to exhibit lower risks of relapse [25,26,29,32,34] compared to the others maintenance protocols.

### 4.6. Maintenance TMS: A New Resource against Recurrent Depression?

While the biological underpinnings underlying magnetic stimulation are not yet fully understood, results from the current systematic review indicate that maintenance TMS could be a globally valuable tool for both MDD and TRD. Despite the heterogeneity, maintenance protocols appear to significantly reduce the relapse rates in responders. In some cases, maintenance TMS has been found to be more effective than oral antidepressant therapy in reducing the risk of relapse, either when used in combination with oral antidepressants or as monotherapy [26].

TMS may also be efficacious in sustaining remission because of its low occurrence of adverse effects, which can amplify treatment fidelity and thus guarantee adequate compliance to treatment. Poor adherence and compliance to oral antidepressant treatment still represent a common issue in the treatment of depression [43], which can lead to a lack of efficacy and an increased risk of relapse or chronicity. While it is generally accepted to discontinue antidepressant therapy after the first depressive episode, typically 6–12 months after remission, the presence of previous depressive episodes represents a strong contraindication to treatment discontinuation. Additionally, discontinuation is often secondary to reported side effects [44] and treatment-emergent emotional blunting related to SSRI or SNRI therapy. On the contrary, TMS administration is entirely dependent on medical staff for dosage and timing, and monthly treatment can overcome compliance issues, consolidating the results achieved with previous acute TMS treatment and preventing relapses. Similarly to long-acting medications in psychotic disorders, which ensure patient compliance and proper dosage control, the use of monthly maintenance TMS protocols in depression could also ensure adequate adherence to treatment and dosage monitoring. In this way, TMS may be used as a depot-like tool to reduce the risk of relapse in depressive disorders, thus representing an innovative way to manage depressed patients.

### 4.7. Limitations

This systematic review shows many limitations. First, the small number of studies performed to date about TMS maintenance protocols makes the results still inconclusive. Among these studies, only three randomized controlled trials [24,25,34] were conducted to investigate maintenance protocols, with the remaining being open-label studies or case reports/series. Additionally, no studies were conducted in the past four years, suggesting a lack of scientific attention to the possibility of sustaining the effectiveness of TMS through multiple applications over time. Furthermore, another significant limitation of the current study is the absence of a meta-analytical approach, which could not be employed due to the limited number of RCTs included.

Finally, the wide heterogeneity in protocols and methods of included studies represents an important limitation to drawing solid conclusions.

## 5. Conclusions

To date, there is not a common consensus about maintenance TMS protocols for MDD and TRD. However, despite a lack of RCT and the heterogeneity of studies in the current literature, maintenance TMS appears to be a resourceful strategy to maintain acute antidepressant treatment effects and significantly reduce the risk of relapses over time.

The ease of administering TMS, the low rates of side effects, and the possibility of monitoring adherence to treatment are important factors to consider when evaluating the future use of maintenance TMS. Therefore, studies that evaluate the long-term effectiveness of maintenance protocols are crucial. Moreover, further studies are needed to better clarify the underlying neuromodulation mechanisms of TMS in order to individuate specific protocols based on neurobiological criteria rather than clinical parameters.

## Figures and Tables

**Figure 1 jpm-13-00697-f001:**
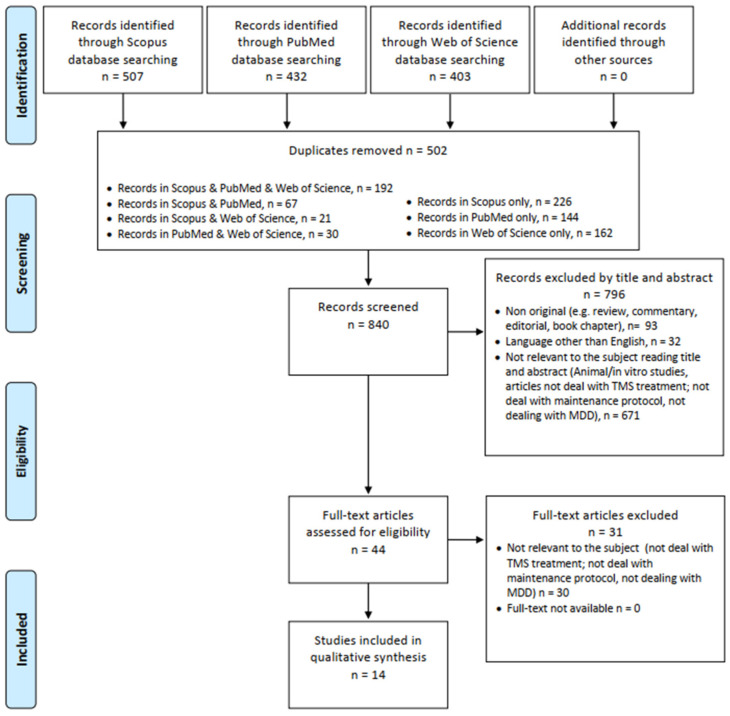
PRISMA flow diagram.

**Table 1 jpm-13-00697-t001:** Acute protocols sorted by study design and alphabetical order.

Reference	Study Design	Sample		Acute Protocol						Distance between Acute and Maintenance Protocol
		Size (n)	Type	Duration (Weeks)	Type of TMS	Coil Position	Frequency; Intensity (% RMT)	Stimulation Time; N of Pulses	N of Total Sessions	
Benadhira et al., 2017 [24]	RCT	58 (35 to maintenance)	TRD	4	rTMS	LDLPFC	10 Hz; 110%	15 min; 2000	20	1 week
Levkovitz et al., 2015 [25]	RCT double-blinded	181: 89 active, 91 sham	TRD	4	dTMS	LDLPFC	15 Hz; 120%	30 min; 980	20	1 week
Wang et al., 2017 [26]	RCT assessor-blinded	281 tot (rTMS = 91, ADP = 108, rTMS + ADP = 82)	MDD	ND	rTMS	LDLPFC	ND	ND	ND	ND
Fitzgerald et al., 2013 [27]	OL	35	26: MDD; 9: BD	ND	rTMS	LDLPFC; RDLPFC; bilateral	10 Hz on LDLPFC at 110% (N = 14); 1–5 Hz on RDLPFC (N = 12); 1–1 Hz Bilateral (N = 6); 10–1 Hz Bilateral (N = 3) at 110%	15 min; 1500 (LDLPFC); 900 (LDLPFC)	ND	1–3 months
Haesebaert et al., 2018 [28]	OL	66 (25 rTMS; 22 Venlafaxine; 19 Combined)	TRD	2 to 6	rTMS	RDLPFC	ND	ND	ND	4 weeks
Harel et al., 2012 [29]	OL	29	MDD	4	rTMS	LDLPFC	20 Hz; 120%	15 min; 1680;	20	1 week
Philip et al., 2016 [30]	OL	49:23 maintenance TMS protocol (SCH group), 26 observation (OBS group)	MDD	6 weeks + 3 weeks of TMS tapering	rTMS	5,5 cm anterior from the MT location, along a left superior oblique plane	10 Hz; 120%	37 min 30 s; 3000	30	4 weeks
Pridmore et al., 2018a [31]	OL	39	TRD	4	rTMS	LDLPFC	10 Hz; 110%	12 min; 3000	40	4 weeks
Pridmore et al., 2018b [32]	OL	14	TRD	4	rTMS	LDLPFC	10 Hz; 110%	12 min; 3000	40	4 weeks
Richieri et al., 2013 [33]	OL	59: 20 weeks of maintenance TMS (n = 37) or no additional TMS treatment (n = 22)	TRD	4	rTMS	LDLPFC-RDLPFC	LDLPFC: 10 Hz; 120% RDLPFC: 1 Hz; 80%	LDLPFC: 20 min; 2.000 RDLPFC: 14 min; 720	20	ND
Yip et al., 2017 [34]	OL double-blinded	33	TRD	4	dTMS	LDLPFC	18 Hz; 120%	20 min; 1980	20	1 week
O’ Reardon et al. 2006 [35]	CS	10	TRD	ND	rTMS	LDLPFC	ND	20 or 30 min; 2000 or 3000	ND	1 week
Chatterjee et al., 2012 [36]	CR	1 (three separate maintenance treatments: a; b; c)	TRD	3	rTMS	LDLPFC	15 Hz; 100%	22 min; 4000	20	a: 6 weeks; b: 7 weeks; c: 12 weeks.
Langguth et al., 2006 [37]	CR	1	MDD	3	rTMS	LDLPFC	20 Hz; 90%	18 min; 2000	15	5 weeks

**Table 2 jpm-13-00697-t002:** Maintenance protocols sorted by study design and alphabetical order.

Reference	Study Design	Sample		Maintenance Protocol				Results
		Size (n)	Type	Duration (weeks) and n of Stimulation/Week	Frequency; Intensity (% RMT)	Stimulation Time; N of Pulses	N of Total Sessions	
Benadhira et al., 2017 [24]	RCT	58 (35 to maintenance)	TRD	44 weeks (11 months); 3/wk for 2 wks; 2/wk for 2 wks; 1/wk for 2 months; 2 per month for 8 months.	10 Hz; 110%	15 min; 2000	34	HDRS: Significant improvement in active group from 1st to 4th month. No difference from 5th month to endpoint. No significant differences in HDRS-17, HAD, BDI and CGI scores at any endpoint. Effect of active treatment could be sustained for two months (no clinical differences at 2nd and 3rd month between groups). Antidepressant effect of maintenance sessions appeared three months after the treatment. One session per week could maintain antidepressant effect. Two sessions/month could be insufficient to maintain antidepressant effect.
Levkovitz et al., 2015) [25]	RCT double-blinded	181:89 active, 91 sham	TRD	2/wk dTMS; 12 weeks following the acute-phase,	18 Hz; 120%	20 min; 1980	24	Response and remission rates were higher in the dTMS than in the sham group (response: 38.4 vs. 21.4%, p50.013; remission: 32.6 vs. 14.6%, *p* = 0.005). These differences between active and sham treatment were stable during the 12-week maintenance phase. From week 5 (end of acute phase) to week 16 (end of mantainance phase) response rates increase from 38.4% to 44.3% in active group, and from 21.3% to 25.6% in the sham group.
Wang et al., 2017 [26]	RCT assessor-blinded	281 tot (rTMS = 91, ADP = 108, rTMS + ADP = 82)	MDD	52 weeks, monthly clustered rTMS maintenance treatment, which involved 10 sessions over a 5-day period for the first 3 months and 5 sessions over a 3-day period thereafter.	10 Hz; 120%; if not tolerated 80%	15 min; 1150	75	rTMS + ADP and rTMS significantly reduced the risk of relapse/recurrence compared with ADP (*p* = 0.000), with hazard ratios of 0.297 and 0.466, respectively. Both rTMS-containing regimens produced significantly lower relapse/recurrence rates than ADP (15.9% and 24.2% vs. 44.4%, *p* < 0.001). TMS + ADP reduce the risk of relapse/recurrence and the relapse/ recurrence rate by 8.3%.
Fitzgerald et al., 2013 [27]	OL	35	26: MDD; 9: BD	Mean duration: 12 ± 9.7 months; 5 stimulations/month	10 Hz on LDLPFC at 110% (N = 14); 1–5 Hz on RDLPFC (N = 12); 1–10 Hz Bilateral. 110%	15 min. N of pulses: 1500 (LDLPFC); 900 (LDLPFC)	5–120	On 25 relapses (mean 10.2 months), 15 withdrew and 10 responded to a new active rTMS protocol followed by maintenance On 10 remissions, 4 withdrew and 6 remained well (mean 12.0 months).
Haesebaert et al., 2018 [28]	OL	66 (25 rTMS; 22 Venlafaxina; 19 Combined)	TRD	52 2/wk for one month; 1/wk for 2 months; once every two weeks for 9 months.	1 Hz; 120%	6 trains of a 1-min duration separated by 30-s inter-train “off” periods; 360	34	Remission rate -rTMS group: 18.7% (n = 3) -venlafaxine group: 35.3% (n = 6) -combination group: 33.3% (n = 4) No difference between the groups regarding the number of patients who maintained remission at the end of the 12-month follow-up (Chi2 = 1.25; *p* = 0.3) Non-relapse rate (HDRS < 15) -rTMS group: 40.0% (10 of 25) -venlafaxine group: 45.1% (10 of 22) (Chi2 = 0.33; *p* = 0.8) -combination group: 36.88% (7 of 19) Relapse rate -rTMS group: 4.0% -venlafaxine group: 4.5% -combination group: 5.3% (Chi2 = 0.04; *p* = 0.9) Probability of non-relapse -rTMS group: 80.0% -venlafaxine group: 59.1% -combination group: 78.9% No significant difference was identified between the three groups regarding the survival distribution using the log rank method (Chi2 = 3.2848; *p* = 0.19).
Harel et al., 2012 [29]	OL	29	MDD	Continuation type I: 8 weeks, twice/week Continuation Type II: 10 weeks, once/week	20 Hz; 120%	14 min; 1680	Continuation type I: 16 sessions Continuation Type II: 10 sessions	Kaplan–Meier estimated probability of response was 46.15% (SE = 9.78%) at the end of the acute phase, and 81.12% (SE = 9.32%) at the end of the study (22 weeks). Probability of remission at the end of the acute phase was 26.92% (SE = 8.70%) and 71.45% (SE = 10.99%) at the end of the study. Response in the acute phase was indicative of response in the continuation phases.
Philip et al., 2016 [30]	OL	49:23 maintenance TMS protocol (SCH group), 26 observation (OBS group)	MDD	40 weeks; once every month	10 Hz; 120%	37 min 30 s; 3000	11	32.7% completed all 53 weeks of the study -no statistically significant group differences in the primary outcome variable (i.e., the number of patients who did not require TMS reintroduction at any observation point during the maintenance phase): 39% in the SCH group and 35% in the OBS group à maintenance TMS schedule of only one treatment per month is not sufficient to prevent return of depressive symptoms within the year
Pridmore et al., 2018a [31]	OL	39	TRD	20 weeks—once every month over 3 or 5 days/week	10 Hz; 110%	12 min; 3000	50	Before TMS series 70% were no longer in remission (being in partial remission or relapse), and after TMS series, 79% were in remission. Pre-maintance HAM-6 = 6.24 ± 2.78; post-maintenance HAM-6 = 3.30 ± 2.28.
Pridmore et al., 2018b [32]	OL	14	TRD	52 weeks—once month over 3 or 5 days/week	10 Hz; 110%	12 min; 3000	at least 60 sessions	12/14 Patients (85%) were on remission during and after the TMS mantainance period. Relapse rate: 15%
Richieri et al., 2013 [33]	OL	59: 20 weeks of maintenance TMS (n = 37) or no additional TMS treatment (n = 22)	TRD	three sessions in week 1, two sessions in weeks 2 and 3, one session per week for 2 weeks one session per 2 weeks for 2 months, and then one session per month for 2 months.	10 Hz; 120% 1 Hz; 80%	LDLPFC: 20 min; 2.000 RDLPFC: 14 min; 720	15	Maintenance TMS was associated with a significantly lower relapse rate (37.8% vs 81.8%) in patients with pharmacoresistant depression in routine practice among responders.
Yip et al., 2017 [34]	OL double-blinded	33	TRD	2/wk dTMS 12 weeks following the acute-phase,	18 Hz; 120%	20 min; 1980	24	24 participants (72.7%) achieved responder status during at least one rating with dTMS continuation—20 (60.6%) within four weeks, with 13 (39.4%) consistently meeting response criteria for the duration of the study. 20 (63.6%) achieved remission status at some point during treatment continuation.
O’ Reardon et al., 2006 [35]	CS	10	TRD	1 or 2 session/weekly for periods ranging from 24 weeks to 288 weeks	10 Hz;100% (n = 9) 20 Hz;100% (n = 1)	20 or 30 min; 2000 or 3000	patients with marked benefits: mean 257 sessions; patients with moderate benefits: mean 125 sessions; patients with minimal benefits: mean 98 sessions	7/10 Subjects experienced marked or moderate benefit. 3 cases were without any oral antidepressant medications
Chatterjee et al., 2012 [36]	CR	1	TRD	a: 1 wk; 5/wk; b: 8 wk; 5/wk; c: 8 wk; 5/wk.	20 Hz; 100%	22 min; 4000	a: 5; b: 20; c: 20.	In all the three mantenaince protocols (a, b, c), remission from the current MDE was achieved (MADRS score: a = 6, b = 4, c = 4)
Langguth et al., 2006 [37]	CR	1	MDD	52 weeks 5 sessions of daily rTMS/monthly	20 Hz; 90%	18 min; 2000	60	HAM-D score pre-TMS = 30 HAM-D score post-TMS: ranged between 0 and 3 every treatment week in 12 months

OL: Open Lable; RCT: Randomized Controlled Trial; CS: Case Series; CR: Case Report; MDD: Major Depressive Disorder; TRD: Treatment-Resistant Depression; R/L-DLPFC: Right/Left Dorsolateral Prefrontal Cortex; rTMS: repetitive TMS; dTMS: deep TMS; RMT: Right Motor Threshold; MADRS: Montgomery Asberg Depression Rating Scale; HDRS-17: Hamilton Depression Rating Scale—17 items; HAD: Hospital Anxiety and Depression; BDI: Beck Depression Inventory; CGI: Clinical Global Impression.

**Table 3 jpm-13-00697-t003:** Type of Maintenance Protocols.

Maintenance Protocols	Description	Included Studies
Tapering rTMS	Referring to a progressive and gradual reduction of session and stimulation across several weeks.	Benhadira et al., 2017 [24]; Haesebaert et al., 2018 [28]; Philip et al., 2016 [30]; Richieri et al., 2013 [33]
Cluster rTMS	5 intensive sessions delivered over 2.5–5 days, separated by monthly or greater non-treatment periods.	Fitzgerald et al., 2013 [27]; Langguth et al., 2006 [37]; Pridmore et al., 2018a [31]; Pridmore et al., 2018b [32]; Wang et al., 2017 [26]
Continuous rTMS	Maintenance sessions are delivered within the first week after acute protocol.	Harel et al., 2012 [29]; Levkovitz et al., 2015 [25]; O’ Reardon et al., 2006 [35]; Yip et al., 2017 [34]
Rescue rTMS	Multiple protocols delivered during a relapse of a depressive episode	Chatterjee et al., 2012 [36]

## Data Availability

Data from the present systematic review are available upon request to the corresponding author.

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
