# Peer review of "Investigating the Role of Maintenance TMS Protocols for Major Depression: Systematic Review and Future Perspectives for Personalized Interventions"

_jpm, 2023, doi:10.3390/jpm13040697_

Round 1

Reviewer 1 Report

I think authors should mention in the section of limitations that a metanalysis was not done. This approach would have increased the value of this study. 

Author Response

Dear Reviewer,

We sincerely appreciate your valuable feedback and insightful review, which have contributed to the enhancement of our manuscript's overall quality. In accordance with your request, we have included the following statement in the limitations section: “Another significant limitation of the current study is the absence of a meta-analytical approach, which could not be employed due to the limited number of RCTs included.”

Reviewer 2 Report

An interesting and factually correct article on rTMS. An important clinical topic, the authors have done their due diligence.
Article worth publishing.

Author Response

Referee #2

Comments to the Author

An interesting and factually correct article on rTMS. An important clinical topic, the authors have done their due diligence. 
Article worth publishing.

Dear Reviewer,

We are delighted that you found our article to be of interest and suitable for publication, and we would like to express our gratitude for your positive remarks.

Reviewer 3 Report

The authors reviewed the current evidence about TMS maintenance protocols for MDD and TRD. There is no  global consensus on how a rTMS maintenance protocol should be carried on these patients. The manuscript is well-written and can fill the research gap in knowledge about TMS maintenance protocols for MDD and TRD.

I have only a few comments listed below to improve the quality of the manuscript. 

Line 268

Target populations

patients who responded to acute TMS treatments patients treated with classical antidepressant therapies non-responders patients.

The authors may have their comment on highly potential target populations for rTMS maintenance. 

Line 278

This apparently discretionary choice could reflect the lack of superiority evidence of LF or HF stimulation protocols over right or left DLPFC in acute Depressive Episodes [38].

LF rTMS is not approved by FDA for treating TRD. In general, the evidence for LF rTMS in treating TRD is generally considered less sufficient than HF rTMS. Can the authors provide more updated literature to support this statement? 

Line 314

4.3 Temporal distance from acute to maintenance protocol 

...As already mentioned before, part of this result could be correlated to “late-responder” phenomenon [19], but could also be seen as an “enhancement” of the acute effects....

Undoubtedly, studies lasting longer than five months can be deemed more reliable and less affected by immediate treatment effects.......

This section is well-written and can help the readers clearly understand the margin between acute treatment and rTMS maintenance. That also guide the design of future RCT on rTMS maintenance for MDD or TRD.

Line 261

Table 4 represents the different maintenance protocols observed in the retrieved studies, which can be resumed in 4 distinct types (Tapering, Cluster, Continuous and Rescue rTMS).

Since I cannot find the Tables in the manuscript, please specify the 4 distinct types. 

In the reviewer’s opinion, the above-mentioned issues need to be addressed by the authors.

Author Response

Referee #3

The authors reviewed the current evidence about TMS maintenance protocols for MDD and TRD. There is no  global consensus on how a rTMS maintenance protocol should be carried on these patients. The manuscript is well-written and can fill the research gap in knowledge about TMS maintenance protocols for MDD and TRD.

I have only a few comments listed below to improve the quality of the manuscript. 

Dear Reviewer,

Thank you very much for the valuable feedback you have provided, which has allowed us to enhance the quality of the manuscript and incorporate crucial information for the reader. We hope that we have addressed your requests in an appropriate manner.

  1. Line 268

Target populations

patients who responded to acute TMS treatments patients treated with classical antidepressant therapies non-responders patients.

The authors may have their comment on highly potential target populations for rTMS maintenance. 

Thank you for the precious comment, which provides us the opportunity to improve the quality of our manuscript. Indeed, offering a deeper understanding of the population type that could benefit most from TMS maintenance protocols is an essential aspect. We have included the following statement in the aforementioned paragraph:

“Despite the considerable heterogeneity in the target population selected for maintenance TMS protocols across various studies, it appears that implementing maintenance protocols in patients who have already responded to acute TMS treatments may exhibit a degree of efficacy. “

  1. Line 278

This apparently discretionary choice could reflect the lack of superiority evidence of LF or HF stimulation protocols over right or left DLPFC in acute Depressive Episodes [38].

LF rTMS is not approved by FDA for treating TRD. In general, the evidence for LF rTMS in treating TRD is generally considered less sufficient than HF rTMS. Can the authors provide more updated literature to support this statement? 

We thank the reviewer for the opportunity to enhance the overall quality of our manuscript. Indeed, although HF rTMS is commonly considered superior to LF rTMS, data from the literature does not support this statement (see: 10.1001/jamapsychiatry.2016.3644, 10.1016/j.psychres.2013.09.007). Specifically, a well-designed recent systematic review with network meta-analysis published in JAMA Psychiatry, encompassing 81 studies, failed to identify any statistically significant difference between HF and LF protocols in depressive disorders. 

We update the references in the aforementioned statement.

  1. Line 314

4.3 Temporal distance from acute to maintenance protocol 

...As already mentioned before, part of this result could be correlated to “late-responder” phenomenon [19], but could also be seen as an “enhancement” of the acute effects....

Undoubtedly, studies lasting longer than five months can be deemed more reliable and less affected by immediate treatment effects.......

This section is well-written and can help the readers clearly understand the margin between acute treatment and rTMS maintenance. That also guide the design of future RCT on rTMS maintenance for MDD or TRD.

Thanks very much for all this kind comment. We are honoured that our work fits your expectations. 

  1. Line 261

Table 4 represents the different maintenance protocols observed in the retrieved studies, which can be resumed in 4 distinct types (Tapering, Cluster, Continuous and Rescue rTMS).

Since I cannot find the Tables in the manuscript, please specify the 4 distinct types. 

We sincerely apologize to the reviewer. During the submission process, as it was not feasible to incorporate the tables within the same document as the manuscript, Table 4, along with the other tables, was provided in a separate document when submitted to the journal.

Here you can find a copy of table 4:

Table 4. Type of Maintenance Protocols.

Maintenance protocols

Description

Included Studies

Tapering rTMS

Referring to a progressive and gradual reduction of session and stimulation across several weeks.

Benhadira et al. 2017; Haesebaert et al. 2018; Philip et al. 2016; Richieri et al. 2013

Cluster rTMS

5 intensive sessions delivered over 2.5-5 days, separated by monthly or greater non-treatment periods.

Fitzgerald et al. 2013; Langguth et al. 2006; Pridmore et al 2018a; Pridmore et al. 2018b; Wang et al. 2017

Continuous rTMS

Maintenance sessions are delivered within the first week after acute protocol.

Harel et al. 2012; Levkovitz et al. 2015; O’ Reardon et al. 2006; Yip et al. 2017

Rescue rTMS

Multiple protocols delivered during a relapse of a depressive episode

Chatterjee et al., 2012
